# Severe Acute SARS-CoV-2 Infection and Long COVID: What Do We Know So Far? New Challenges in Diagnosis and Management

**DOI:** 10.3390/diseases13100337

**Published:** 2025-10-13

**Authors:** Sara Mazzanti, Francesco Barchiesi, Francesco Pallotta, Ilenia Luchetti, Andrea Giacometti, Lucia Brescini

**Affiliations:** 1AUO delle Marche, Università Politecnica delle Marche, Via Conca, 60100 Ancona, Italy; pallottafrancesco1993@gmail.com (F.P.); ilenialuchetti@hotmail.it (I.L.); a.giacometti@staff.univpm.it (A.G.); l.brescini@staff.univpm.it (L.B.); 2A.O. Ospedali Riuniti Marche Nord, Università Politecnica delle Marche, Via Cesare Lombroso, 1, 61122 Pesaro, Italy; f.barchiesi@staff.univpm.it

**Keywords:** long COVID, inflammation, pneumonia

## Abstract

Background/Objectives: The long-term impact of the COVID-19 pandemic is not just limited to socioeconomic aspects; there are also important health issues to consider. Among these, one of the most important and obvious is long COVID. Despite a significant amount of scientific work having been published, this condition is still semi-unknown. The objective of this study was to collect useful information for the clarification of some epidemiological, clinical, and laboratory characteristics of this disease. Methods: This was a single-center study carried out at the Infectious Diseases Clinic of the hospital “AUO delle Marche” on all patients hospitalized for COVID-19 between November 2021 and March 2022. Results: From the data, it emerged that, following the resolution of the acute phase of SARS-CoV-2 infection, the majority of people experienced health problems that persisted for at least 6 months. The manifestations and outcomes affect different systems; therefore, long COVID, like COVID-19, has systemic involvement and the clinical manifestations may be residues of the damage caused by the disease during the acute phase, or new manifestations whose pathogenesis is still a matter of discussion. Conclusions: The persistence of inflammation and the dysregulation of the immune system represent some of the pathogenetic hypotheses. Inflammation could therefore represent one of the physiopathogenetic mechanisms of long COVID, and it is possible that it is responsible for the clinical symptoms that appear in the months following the resolution of the acute phase of the disease.

## 1. Introduction

Long COVID is a condition involving the persistence of often severe symptoms following SARS-CoV-2 infection [1]. The WHO defines this syndrome as a continuation or development of new symptoms 3 months after the initial SARS-CoV-2 infection, with these symptoms lasting for at least 2 months with no other explanation. Several studies show that long COVID can occur after acute infections that may or may not have been serious enough to require hospitalization [2].

The impact of long COVID on the population is evident and the syndrome is recognized as a clinical entity; several studies are underway to better define its characteristics, starting from the causes.

The symptoms of long COVID can vary from person to person; in general, they include persistent fatigue, tiredness, weakness, muscle and joint pain, and lack of appetite.

The specific symptoms manifest themselves at the respiratory, cardiovascular, neurological, gastrointestinal, and psychological levels [3,4,5]. Examples include dyspnea, persistent cough, chest pain and feelings of oppression, tachycardia and palpitations, and changes in blood pressure. Headaches; difficulty concentrating and remembering (the so-called “brain fog”); disorders of smell, taste, and hearing; nausea and vomiting; loss of appetite; abdominal pain; diarrhea; and gastroesophageal reflux are also reported as symptoms. The dermatological system can be involved, including instances of rash, alopecia, and telogen effluvium [6,7]. Sleep disturbances, depressed mood (sadness, irritability, intolerance, lack of interest in previously enjoyed activities), anxiety, stress, and psychosis are other symptoms related to the central nervous system [8].

The pathogenic mechanism of long COVID is not known with certainty, but many argue that it may be caused by the persistence of an inflammatory state, with the production of cytokines even after the acute phase of the disease [9,10,11]. It is even more difficult to establish the etiology of this condition when it occurs after an acute form of severe COVID-19 infection (e.g., severe pneumonia that required ventilation). In these cases, in fact, there are also organic alterations that can contribute to the persistence of symptoms or discomfort [12].

Regarding epidemiology, some studies have demonstrated that there are some predisposing conditions for developing long COVID, such as gender (women aged 35–50 years), disadvantaged socioeconomic conditions, comorbidities (type 2 diabetes, BPCO, chronic lung disease, heart failure, chronic kidney disease, and obesity), more severe acute illness, and being unvaccinated [13]. This condition occurs in a high percentage of patients, and this determines a significant impact both on the national health system and on the socioeconomic system of the country.

In this scenario, the first aim of this study was to evaluate the long-term outcomes and complications of coronavirus disease (COVID-19) in patients admitted to the Infectious Diseases Clinic of the hospital “AUO delle Marche”. Follow-up was carried out one month, three months, and six months after hospital discharge.

The second outcome is to contribute to the definition and understanding of long COVID syndrome, in particular by looking for possible laboratory markers in the prediction of the disease.

## 2. Materials and Methods

Study design

This prospective and monocentric study was conducted by enrolling COVID-19 patients hospitalized in the Infectious Diseases Clinic department. Participants were followed prospectively in the period March 2020–February 2022 through outpatient visits carried out one month, three months, and six months after hospital discharge.

Sample and setting

During the outpatient visit, a medical doctor investigated the presence of new clinical symptoms, symptoms that were present before the SARS-CoV-2 infection that worsened during the disease, or symptoms that persisted from the hospitalization period.

The following examinations were performed:
−General objective examination, with a focus on the objective examination of the chest;−Neurological examination;−Measurement of blood pressure, heart rate and oxygen saturation;−Measurement of body weight;−Lung ultrasound: the presence of A lines corresponds to a picture of healing while the detection of multifocal, separate or confluent B lines or consolidations indicates the persistence of inflammation;−Evaluation of lung imaging through CT or X-ray reports performed at the Ancona hospital or in other facilities;−Evaluation of blood chemistry tests performed at the Ancona hospital or in the laboratories of other facilities;−Consultation of reports regarding pathologies of different specialist relevance such as dermatological visit, physiatric visit, ENT visit, ophthalmological visit;−Consultation of spirometry and Diffusing Capacity of the Lungs for Carbon Monoxide (DLCO) reports; these examination were performed by local pneumologist according to the guidelines of American Thoracic Society and European Respiratory Society Technical Statement [14];−Consultation of cardiological visit reports.

Inclusion criteria

Inclusion criteria were as follows: having been hospitalized for SARS-CoV-2 pneumonia (all patients at the time of admission to the hospital performed an oro-nasopharyngeal swab to search for the viral genome, with the rRT-PCR technique, and received positive results, and they all undertook a chest CT or chest X-ray that showed findings compatible with interstitial pneumonia of viral etiology) with severe respiratory failure requiring continuous positive airway pressure (cPAP) or intubation, and having been followed up at 6 months.

Exclusion criteria

Exclusion criteria were as follows: having been hospitalized for mild infection without signs of pneumonia and failure to carry out follow-up visits.

Statistical analysis

An initial descriptive analysis to arrive at the results relating to the first outcome was performed directly via Microsoft Excel. The data were subsequently analyzed with SPSS 30.0.0 software to allow the comparisons necessary for the second outcome. Before transferring the data obtained from Excel to the statistical analysis software, the latter were transformed into nominal dummy qualitative variables (dichotomous), where there were continuous quantitative data such as in laboratory tests, and cut-offs were chosen that allowed two categories to be assigned to each variable. With regard to the second outcome, the dependence or independence of laboratory anomalies and the presence of long COVID at 3 months for each symptom investigated was assessed. After having descriptively represented these associations in some contingency tables, the chi-square test was chosen as the most suitable and simple method for undertaking multiple univariate analysis of these dichotomous variables.

## 3. Results

A total of 191 patients were recruited through the post COVID-19 Diagnostic Therapeutic Care Pathway (PDTA) of the Infectious Diseases Clinic of Ancona; of these, 188 were evaluated after one month from discharge, 158 after three months and 103 after 6 months. Some participants concluded the follow-up early due to good general health conditions.

The median age of the sample examined is 61 years with a prevalence of men compared to women (58% vs. 42%). Table 1 presents the main clinical characteristics at baseline.

### 3.1. First Outcome

One month after hospital discharge, the most frequently observed symptoms were dyspnea on exertion and asthenia, reported in 43% and 35% of cases, respectively.

The following symptoms in terms of frequency were myalgia (14%), arthralgia (13%), gastrointestinal disorders (12%), cough (9%), skin changes (9%), psychiatric changes (7%), paresthesia (6%), cognitive changes (5%), chest pain (5%), and ageusia and anosmia (5%).

With a prevalence of <5%, the following were reported: hearing changes (4%), tremors (4%), blood pressure changes (3%), vertigo (3%), telogen effluvium (2%), anosmia (2%), headache (2%), palpitations (2%) and decreased vision (1%) (see Figure 1).

With regard to laboratory values, one month after discharge, abnormal values of red blood cells were found in 41% of the cases, Hb in 23% and white blood cells in 11%, and platelet values were altered at a lower percentage, in 6% of cases. The study of inflammation markers showed values of IL-6 > 5.20 pg/mL in 54% of cases, C-reactive protein (PCR) > 0.6 mg/dL in 20% of cases, D-dimer > 230 ng/mL in 31% of cases and fibrinogen > 400 mg/dL in 83% of cases. Regarding the lipid profile, 58% of the analyzed participants had total cholesterol greater than 200 mg/dL, LDL was >116 mg/dL in 54% of cases, HDL < 40 mg/dL in males and <50 mg/dL in females in 20% of cases, triglycerides < 150 in 42% and blood sugar > 110 mg/dL in 22% of cases. Alanine transaminase was >40 mU/mL in 33% of cases (Table 2).

During the outpatient visit, lung ultrasound was performed on 163 patients, of whom 62%, or 101 patients, had B lines.

Fifty patients underwent chest X-ray, highlighting the following:−In 20 (40%), the persistence of the accentuation of the broncho-interstitial pattern;−In 6 (12%), dysventilatory streaks;−In 24 (48%), almost complete resolution of parenchymal thickening and reduction in broncho-interstitial network thickening.

Additionally, 67 patients underwent chest CT, and the images were compared with imaging performed during the hospitalization period.

The observed radiological pictures showed the following:−In 20 cases (30%), almost complete resolution of the inflammatory picture;−In 36 cases (54%), a situation of slight improvement, with reduction in the consolidation aspects but permanence of the GG areas, or a stationary picture;−In 11 cases (16%), dysventilatory fibrotic streaks and micronodularity.

Three months after discharge, the most frequently observed symptoms were dyspnea on exertion and asthenia, reported in 21% and 18% of cases, respectively.

The next most frequent symptoms were myalgia (11%), psychiatric alterations (11%), telogen effluvium (11%), arthralgia (9%), cognitive alterations (9%), gastrointestinal disorders (7%), paresthesia (7%) and skin alterations (7%). With a prevalence of less than 5%, the following were reported: cough (4%), hearing alterations (3%), dizziness (3%), chest pain (1%), tremors (1%), blood pressure changes (1%), headache (1%), palpitations (1%) and decreased vision (1%) (Figure 2).

The analysis of the blood tests, reported in Table 3 below, show that the reduction in red blood cells and Hb, the high inflammation indices and the alteration of the lipid profile persist.

During the outpatient visit, lung ultrasound was performed in 142 patients, and 74 (52%) had pathological B lines.

Six patients underwent chest X-ray, which showed the following:−Complete resolution in 3 patients (50%);−Diffuse accentuation of the broncho-interstitial pattern in 2 patients (33%);−Accentuation of the pulmonary interstitium with the presence of dysventilatory/dysalectatic streaks in 1 patient (17%).

Additionally, 27 patients underwent chest CT, which showed the following:−Complete resolution in 4 patients (19%);−Slight improvement with reduction in the areas of consolidation but permanence of the ground-glass areas in 13 patients (48%);−Dysventilatory streaks and micronodularity in 9 patients (33%);

Six months after discharge, the most frequent symptoms were dyspnea on exertion (14%) and asthenia (10%). The next most frequent were myalgia (8%), arthralgia (8%), psychiatric disorders (8%), gastrointestinal disorders (8%), dizziness (7%), paresthesia (7%), skin alterations (7%), cognitive alterations (7%), hearing alterations (6%), and telogen effluvium (6%), with a 3% prevalence of decreased vision, 2% of cough and 1% of anosmia, headache and palpitations (Figure 3).

At the end of the follow-up, as in the previous months, a reduction in erythrocytes and Hb, persistence of high inflammation indices and alteration of the lipid profile were observed (Table 4).

Comparing the clinical data regarding the entire observation period, we can see that most of the symptoms are present after one month from discharge and tend to decrease over time. Paresthesia, on the other hand, occurs almost constantly during all 6 months, while some symptoms have a later onset time; hair loss, psychic alterations and cognitive alterations were reported with greater frequency in the third month of follow-up, while hearing alterations, vertigo and decreased vision were reported in the sixth month (Figure 4).

During the follow-up, the evaluation of functionality was performed by spirometry and alveolar–capillary diffusion of carbon monoxide (DLCO).

The reports of 11 patients showed restrictive ventilatory deficit, while those of 18 patients showed a reduction in DLCO (Figure 5).

Overall, 96 participants underwent a cardiology examination, which showed normal results in 55 patients. Among the reports that found pathologies, 16 showed pre-existing pathologies such as patent foramen ovale, aortic aneurysm, valvular insufficiency or stenosis and chronic ischemic heart disease; there were 10 arrhythmias, including atrial fibrillation, extrasystoles, sinus excitation disorders and first-degree atrioventricular block; 7 diastolic dysfunctions; 6 instances of hypertensive heart disease; and 2 outcomes of myopericarditis (Figure 6).

### 3.2. Second Outcome

Table 5 and Table 6 show the attempt to identify, among the laboratory markers, those that could be candidates to become markers of long COVID syndrome.

The two symptoms most reported by patients were taken into consideration: dyspnea and asthenia. The data was collected at the visit in the 3rd month post-COVID, as it represented the moment at which the data was most numerous.

The two groups of patients who did and did not present the related long COVID symptoms were then compared to look for any statistically significant differences that would allow the laboratory data to be scientifically correlated with the syndrome.

As regards dyspnea, no variable showed significant difference in the two groups. In the case of asthenia, the presence of leucocytosis and altered ALT at 3 months was found to be statistically related to the presence of the symptom.

## 4. Discussion

The results show that lot of patients present symptoms up to six months after hospital discharge, and the clinical manifestations affect different organs and systems, compromising both physical and psychological aspects. Respiratory, cardiovascular, psychiatric, neurological, gastrointestinal, musculoskeletal and dermatological symptoms emerged, which manifested themselves individually or in combination.

The symptoms reported most frequently throughout the observation period were dyspnea, asthenia, myalgia and arthralgia. The percentage of patients presenting these symptoms, although decreasing during the follow-up, remained the highest throughout the observation period, thus making it possible to define these as the main symptoms of long COVID.

Dyspnea on exertion, and coughing, sequelae of interstitial pneumonia from COVID-19, tended to decrease over time.

At the 6th month after discharge, there was persistence of abnormalities in lung function tests, and in radiological examinations, a third of patients demonstrated consequences of interstitial anomalies or pulmonary vascular anomalies caused by SARS-CoV-2 infection.

Data about the percentage of patients with fibrodisventilatory striae, micronodularity, bronchiectasis and traction bronchiolectasis were detected at the radiological examinations, making it evident that interstitial pneumonia from COVID-19 can determine the development of fibrosis and therefore potentially progressive damage [15,16,17,18].

Among the cardiac problems, all the patterns identified (diastolic dysfunction, hypertensive heart disease and arrhythmias) can be related to viral trigger. Arrhythmias, in particular, are likely related to inflammatory outcomes [19].

The involvement of the central and peripheral nervous system is demonstrated by the presence of ageusia, anosmia, paresthesias, cognitive disorders, headache, dizziness, hypovision and hearing loss.

Ageusia and anosmia may occur during the acute phase of the disease and persist up to 6 months after resolution of the infection.

Paresthesias, referred to as a tingling sensation, was approximately constant throughout the observation period and mainly affected the lower and upper limbs. Cognitive deterioration, so-called “brain fog”, manifests itself with memory deficits, difficulty paying attention and concentrating; these symptoms occurred more frequently three months after discharge and can compromise daily and work activities [20]. Participants also described a feeling of vertigo, tinnitus or dizziness, sometimes in combination with balance problems. This symptomatology peaked at the end of the follow-up, probably indicating a later damage [21].

The skin involvement appears to be more frequent during the first month, and then remained constant in the following months; the skin alterations found were acne lesions, dry skin, itchy rash, vesicular rash, folliculitis, skin xerosis, urticarial lesions and papular lesions, some of which have been defined by dermatologists as probably of post-infectious origin [22].

COVID-19, in addition to physical repercussions, has also led to psychological consequences, among which the following have been diagnosed: anxiety–depressive syndrome, insomnia, mood swings, crying spells and nonspecific maladjustment syndrome. These can significantly compromise the patient’s quality of life. The pandemic context has also played a fundamental role, as the reduction in social relationships, isolation from loved ones and fear of the disease have influenced the psychological component [23,24].

Another manifestation, of dermatological relevance but which can have an impact on the psychological sphere of the patient, is telogen effluvium, or hair loss. It occurred more frequently in the female sex, with a greater prevalence from the third month after discharge, remaining high until the end of the follow-up. These data agree with other groups of patients in the literature [25].

The descriptive analysis of laboratory data highlighted high inflammation parameters in a significant percentage of patients, indices that remained high throughout the observation period. In the part of the analysis relating to the search for markers, it emerged that white blood cells and AST correlate with the presence of asthenia, which is the main symptom attributable to long COVID. This suggests that these parameters are correlated with long COVID and could be considered markers [26,27].

To confirm this hypothesis, a control group of asymptomatic patients would be necessary. In general, this work opens the way for further new studies.

Laboratory tests also show a reduction in red blood cells and hemoglobin, and therefore a condition of anemia, which given the presence of high inflammation indices, can be assumed to be anemia of chronic disease (ACD) [28].

The alteration of lipid metabolism could be a consequence of systemic inflammation and endothelial damage caused by the infection. In the data analysis, patients with LDL values lower than 116 mg/dL were considered non-pathological; however, optimal LDL values depend on the risk factors and comorbidities present, such as diabetes, arterial hypertension and smoking habit, and therefore the number of participants with non-optimal LDL was probably underestimated.

The altered lipid profile observed in some of the patients in this study, associated with the increased incidence rate of cardiovascular diseases in the post-COVID-19 period compared to the pre-exposure period reported in the literature [29], suggests an increase in cardiovascular risk in the period following SARS-CoV-2 infection.

It is known that patients in the acute phase of SARS-CoV-2 infection may have alterations in glycemia and insulin resistance; an Italian multicenter study has demonstrated the persistence of hyperglycemia of varying severity even after recovery from the infection, data in line with the results of this study [30].

This study first of all confirms what has been demonstrated by many other studies in the literature. In particular, it confirms that long COVID represents an emerging public health problem and that the economic impact on the health system is significant. The clinical manifestations highlighted are in line with what has been reported in other studies [31,32]. What the studies on long COVID in recent years have focused on is the pathogenesis of the disease [33,34]. Despite many works having been undertaken, this is still unknown, although other authors have also hypothesized inflammatory, autoimmune, immunomodulated causes, etc. It is known that the development of the disease is linked to individual factors (social, genetic, hormonal, etc.) [35,36].

The search for biomarkers is fundamental for this purpose, and also with a view to new therapies, and it is in this area above all that this work brings innovation and evidence.

This study has several limitations. The fact that all patients were hospitalized for COVID-19 could have caused a longer convalescence. In this sense, it is possible that some symptoms could be due to prolonged immobilization, adverse effects from polypharmacy, etc.

Other possible biases are the advanced average age and the prevalence of male patients in the sample.

## 5. Conclusions

Following the resolution of the acute phase of SARS-CoV-2 infection and a negative swab result, therefore indicating the elimination of the virus from the body, a portion of people who have contracted COVID-19 present health problems that persist for at least 6 months. The manifestations and outcomes affect different systems; therefore, long COVID, like COVID-19, has a systemic involvement and the clinical manifestations may be residues of the damage caused by the disease during the acute phase, or new manifestations whose pathogenesis is still a matter of discussion. The persistence of inflammation and the dysregulation of the immune system represent some of the pathogenetic hypotheses.

Inflammation could therefore represent one of the physiopathogenetic mechanisms of long COVID, and it is possible that it is responsible for the clinical symptoms that appear in the months following the resolution of the acute phase of the disease.

This result opens the way to a possible search for therapeutic strategies for this condition that could be based on drugs that block inflammation in multiple metabolic pathways. Obviously, further studies on larger samples and over a longer follow-up period are needed.

## Figures and Tables

**Figure 1 diseases-13-00337-f001:**
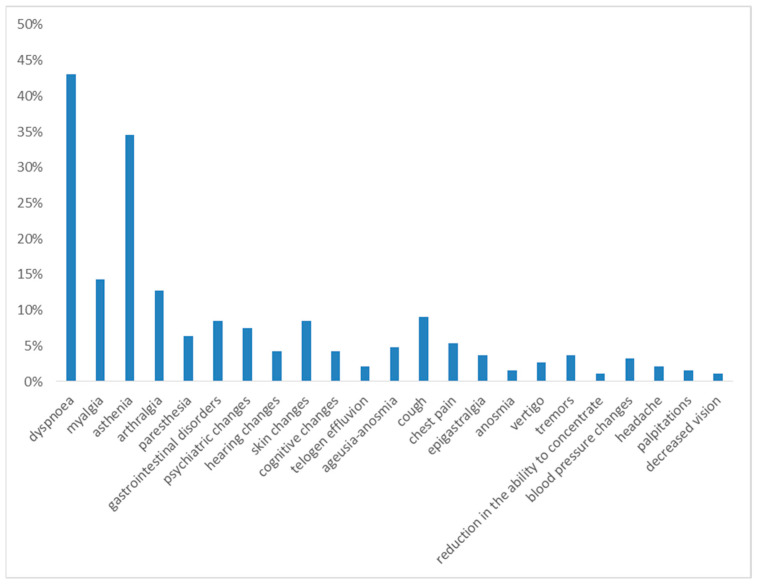
Symptoms at 1st month of follow-up.

**Figure 2 diseases-13-00337-f002:**
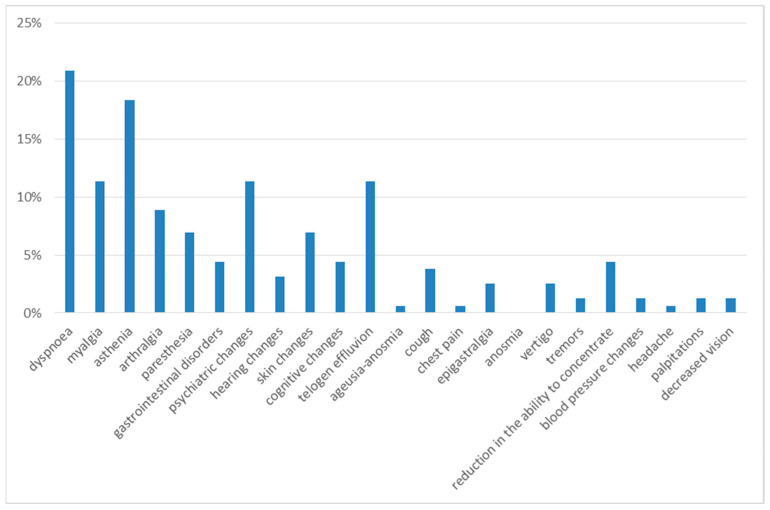
Symptoms at 3rd month of follow-up.

**Figure 3 diseases-13-00337-f003:**
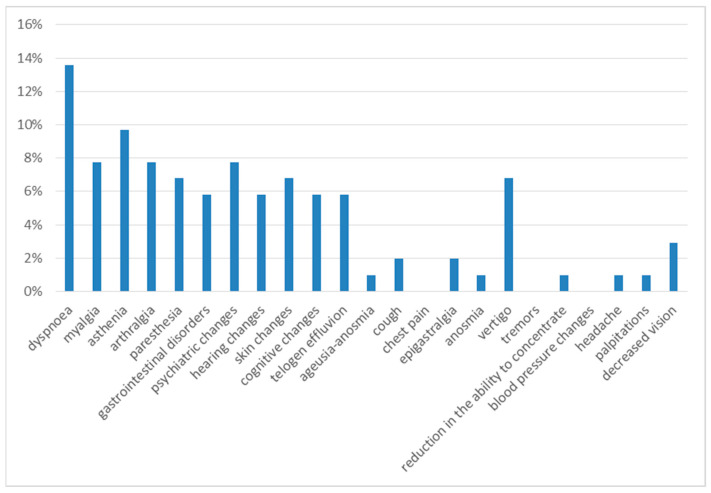
Symptoms at 6th month of follow-up.

**Figure 4 diseases-13-00337-f004:**
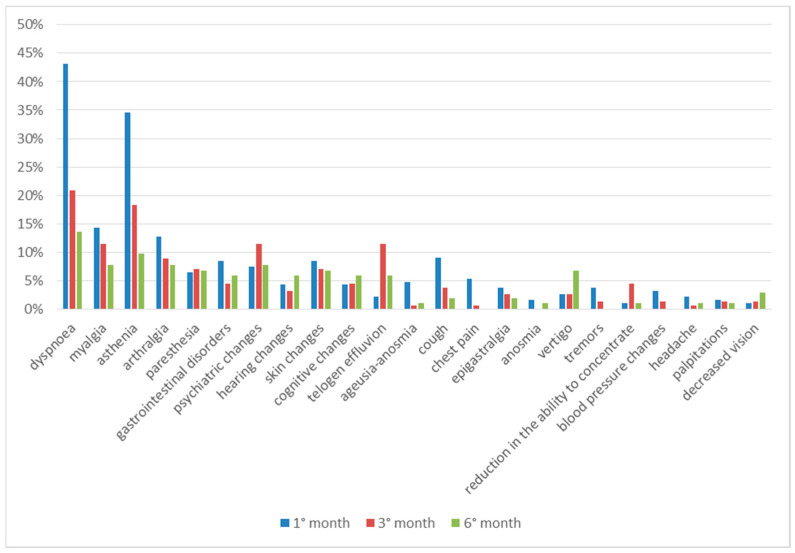
Symptoms in total follow up.

**Figure 5 diseases-13-00337-f005:**
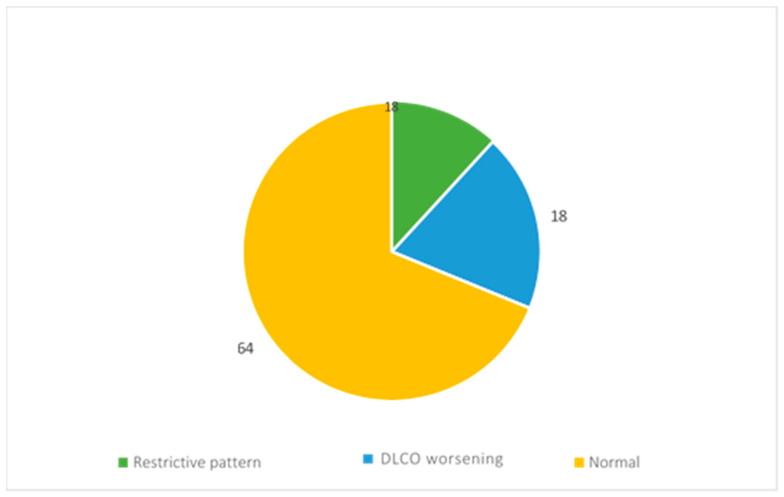
Respiratory function assessment results.

**Figure 6 diseases-13-00337-f006:**
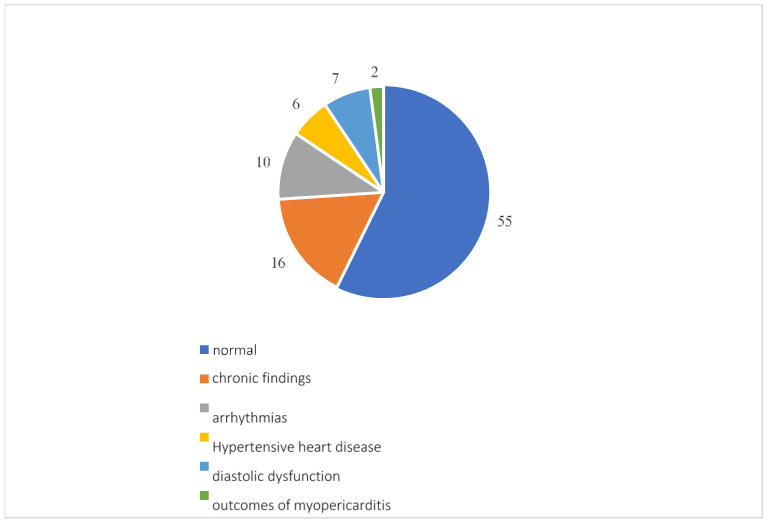
Cardiological examination results.

**Table 1 diseases-13-00337-t001:** Characteristics of patients (*n*: 191).

Characteristics of Patients	*n* (*%*)
Male, *n* (%)	111 (58)
Age, *median (IQR)*	61 (50–71)
BMI, *median (IQR)*	25 (23–28)
Acute comorbidities	
Arrhythmia, *n* (%)	8 (4)
Ischemia, *n* (%)	2 (1)
Bacterial pneumonia, *n* (%)	4 (2)
Pulmonary embolism, *n* (%)	11 (6)
TVP (deep vein thrombosis), *n* (%)	5 (2)
Bacterial coinfection, *n* (%)	4 (2)
ARDS, *n* (%)	17 (9)
Anemia, *n* (%)	74 (39)
Coagulation abnormalities, *n* (%)	29 (15)
Pneumothorax, *n* (%)	4 (2)
Myositis, *n* (%)	2 (1)
Pleural effusion, *n* (%)	15 (8)
Acute kidney failure, *n* (%)	15 (8)
Pancreatitis, *n* (%)	11 (6)
Meningitis, *n* (%)	2 (1)
Liver disease, *n* (%)	65 (34)
Heart failure, *n* (%)	8 (4)
Confusion, *n* (%)	17 (9)
Pericarditis, *n* (%)	2 (1)
Coinfection in other sides, *n* (%)	32 (17)
Septic shock *n* (%)	6 (3)
Rash, *n* (%)	4 (2)
Chronic comorbidities, *n* (%)	
Previous hospitalization, *n* (%)	40 (21)
Malnutrition, *n* (%)	10 (5)
Previous pneumonia (previous 5 years), *n* (%)	10 (5)
BPCO, *n* (%)	15 (8)
HIV, *n* (%)	2 (1)
Hypertension, *n* (%)	84 (44)
Cardiac diseases, *n* (%)	38 (20)
Diabetes mellitus, *n* (%)	27 (14)
Dyslipidemia, *n* (%)	36 (19)
Neurological diseases, *n* (%)	32 (13)
Dialysis, *n* (%)	2 (1)
Solid tumors *n* (%)	6 (3)
Hematological malignancies, *n* (%)	2 (1)
Chronic kidney failure, *n* (%)	13 (7)
Liver chronic diseases, *n* (%)	6 (3)
Asplenia, *n* (%)	2 (1)
Solid organ transplant, *n* (%)	2 (1)

**Table 2 diseases-13-00337-t002:** Blood tests at 1st month of follow-up.

Parameters	Tested Patients (*n*)	Patients with Laboratory Abnormalities (*n*)	%
Leucocyte 10^3^/µL	160	18	11%
Erythrocyte 10^6^/µL	163	67	41%
Hemoglobin g/dL	161	37	23%
Platelets 10^3^/µL	159	10	6%
PCR mg/dL	159	32	20%
D-dimer ng/mL	151	47	31%
Interleukine-6 pg/mL	138	75	54%
Fibrinogen mg/dL	59	49	83%
ALT U/L	159	53	33%
Total cholesterol mg/dL	152	88	58%
Triglycerides mg/dL	149	62	42%
Glycemia mg/dL	136	30	22%
HDL mg/dL	70	14	20%
LDL mg/dL	61	33	54%
Electrophoresis	80	11	14%
LDH U/L	141	19	13%
CPK U/L	154	7	5%

**Table 3 diseases-13-00337-t003:** Blood tests at 3rd month of follow-up.

Parameters	Tested Patients (*n*)	Patients with Laboratory Abnormalities (*n*)	%
Leucocyte 10^3^/µL	139	8	6%
Erythrocyte 10^6^/µL	139	29	21%
Hb g/dL	138	19	14%
Platelets 10^3^/µL	128	12	9%
PCR mg/dL	125	26	21%
D-dimer ng/mL	61	25	41%
IL-6 pg/mL	102	58	57%
Fibrinogen mg/dL	5	3	60%
ALT U/L	122	19	16%
Total cholesterol mg/dL	115	54	47%
Triglycerides mg/dL	112	28	25%
Glycemia mg/dL	32	11	34%
HDL mg/dL	70	19	27%
LDL mg/dL	53	29	55%
Electrophoresis	78	7	9%
LDH U/L	120	31	26%
CPK U/L	122	18	15%

**Table 4 diseases-13-00337-t004:** Blood tests at 6th month of follow-up.

Parameter	Tested Patients (*n*)	Patients with Laboratory Abnormalities (*n*)	%
Leucocyte 10^3^/µL	79	5	6%
Erythrocyte 10^6^/µL	80	21	26%
Hb g/dL	80	10	13%
Platelets 10^3^/µL	77	10	13%
PCR mg/dL	78	17	22%
D-dimer ng/mL	47	20	43%
IL-6 pg/mL	64	23	36%
Fibrinogen mg/dL	1	0	0%
ALT U/L	82	10	8%
Total cholesterol mg/dL	58	23	40%
Triglycerides mg/dL	59	18	31%
Glycemia mg/dL	13	5	38%
HDL mg/dL	33	9	27%
LDL mg/dL	24	12	50%
Electrophoresis	35	3	9%
LDH U/L	72	23	32%
CPK U/L	NA	NA	NA

**Table 5 diseases-13-00337-t005:** Correlation between asthenia and laboratory markers.

	Patients with Asthenia (28)	Patients Without Asthenia (115)	*p*
White blood cells > 8000 (*n*,%)	4 (14)	8 (7)	0.01
PCR > 5 (*n*,%)	8 (28)	30 (26)	0.9
LDH > 300 (*n*,%)	3 (11)	21 (18)	0.4
CPK > 300 (*n*,%)	4 (14)	12 (10)	0.06
ALT > 40 (*n*,%)	9 (32)	12 (10)	0.01
IL-6 > 10 (*n*,%)	2 (7)	24 (20)	0.08

**Table 6 diseases-13-00337-t006:** Correlation between dyspnea and laboratory markers.

	Patients with Dyspnea (32)	Patients Without Dyspnea (110)	*p*
White blood cells > 8000 (*n*,%)	1 (3)	11 (10)	0.5
PCR > 5 (*n*,%)	5 (16)	34 (30)	0.2
LDH > 300 (*n*,%)	6 (19)	18 (16)	0.09
CPK > 300 (*n*,%)	4 (14,2)	12 (11)	0.8
ALT > 40 (*n*,%)	4 (14)	18 (16)	0.9
IL-6 > 10 (*n*,%)	7 (25)	19 (16)	0.5

## Data Availability

Data were collected from the medical case sheets and the laboratory and radiology data, available on the hospital’s electronic database.

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
