# Peer review of "Severe Acute SARS-CoV-2 Infection and Long COVID: What Do We Know So Far? New Challenges in Diagnosis and Management"

_diseases, 2025, doi:10.3390/diseases13100337_

Round 1

Reviewer 1 Report

Comments and Suggestions for Authors

Dear Auhtors,

The article submitted to Diseases is an original article.

Here below my comments/suggestions:

_ You should define any acronyms or abbreviations you use (i.e CPAP, ENT, PCR, DLCO etc..)

_Indicate the guidelines followed for performing spirometry, and specify the blood tests that were carried out.

-Please specify which statistical tests were used and indicate the version of SPSS employed in the analysis

-Was the sample size calculated? If so, please describe the method used for the calculation

-Having a study that evaluates different symptoms, functional variations, and biologial/ functional markers at different times from diagnosis, I would suggest making comparisons of the variables at the different time points. For example (lines 194-196 and Figure 4), some results will certainly be statistically significant. In my opinion, it would be important to have this information, as it could definitely enrich the work.

-Add a table with the baseline characteristics of the patients. In my opinion, knowing the patients’ comorbidities is essential

-Revise the discussion in light of any new results obtained

-line 77 The subject is missing in the sentence

Author Response

Comment 1: You should define any acronyms or abbreviations you use (i.e CPAP, ENT, PCR, DLCO etc..)

Response 1: Done.

Comment 2: Indicate the guidelines followed for performing spirometry, and specify the blood tests that were carried out.

Response 2: Done.

Comment 3: Please specify which statistical tests were used and indicate the version of SPSS employed in the analysis

Response 3: the SPSS version is now specified. The statistical test was just described (chi square).

Comment 4: Was the sample size calculated? If so, please describe the method used for the calculation

Response 4: No, we haven't calculated the sample size.

Comment 5: Having a study that evaluates different symptoms, functional variations, and biologial/ functional markers at different times from diagnosis, I would suggest making comparisons of the variables at the different time points. For example (lines 194-196 and Figure 4), some results will certainly be statistically significant. In my opinion, it would be important to have this information, as it could definitely enrich the work.

Response 5: Thanks, It is an interesting advice and It will be done, in particular a new comparison with variables also at 1 year and over. It has not been done in this work because it is already a very large descriptive analysis and the intent was to give a view of everything without going into detail with statistical correlation

Comment 6: Add a table with the baseline characteristics of the patients. In my opinion, knowing the patients’ comorbidities is essential

Response 6: done.

Comment 7: Revise the discussion in light of any new results obtained

Response 7: done.

Comment 8:line 77 The subject is missing in the sentence

Response 8: Done.

Reviewer 2 Report

Comments and Suggestions for Authors

Here, Mazzanti et al. present a single-centre prospective observational study investigates the long-term effects of severe SARS-CoV-2 infection in patients who were hospitalised with COVID-19 in AUO delle Marche hospital. With 191 patients in the study, the data here is well supported by large numbers.

Clinical, laboratory, and imaging data were collected at 1, 3, and 6 months post-hospital discharge to monitor ongoing and emerging symptoms and evaluate potential biomarkers. Notably, elevated white blood cell count and ALT levels were significantly associated with post-COVID asthenia at 3 months, although the benefits of these are not entirely clear predictive biomarkers.

Minor issues which needs solving:

  • Change the title: the title sounds like a review not a research article
  • Did any patients recover and have no symptom over the 6 month time period?
  • Were the patients vaccinated? It is worth noting that vaccines also appear to protect against long term symptoms of COVID-19 (DOIs: 10.1038/s41541-022-00526-5, 10.1093/cid/ciac630, 10.1093/ofid/ofac464, 10.1038/s41591-022-01840-0) and this should be mentioned.
  • This study also this needs patient information, consider Table S1 in 10.1016/j.immuni.2021.05.010 as a good way to set this out
  • The fact that the cohort were all hospitalised is a key limitation which should be mentioned in the discussion
  • Line 111 – “Some participants concluded the follow up early due to good general health conditions” – does this mean some reported no symptoms at that time point?
  • Bias towards men and older population should be mentioned in the discussion
  • I think it is best to mention that the biomarkers correlate with specific symptoms, not Long COVID in general. In order for the biomarkers to work for Long COVID, a non-symptomatic control is needed in this study.

Major issue:

What this article desperately needs is more connections to the vast Long COVID literature, and discussion of how this works fits into the field as a whole. Many studies have tracked long COVID symptoms over time and the authors should discuss how this work connects to other work in the literature. Additionally, there are several different methods for

Author Response

Comment 1: Change the title: the title sounds like a review not a research article

Response 1: Thank you for the advice. I haven't modified the title because I think it contains general information of the subject but the details of the study project and research are clear in the abstract.

Comment 2: Did any patients recover and have no symptom over the 6 month time period?

Response 2: No. every patients had at least one symptom over the study period.

Comment 3:Were the patients vaccinated? It is worth noting that vaccines also appear to protect against long term symptoms of COVID-19 (DOIs: 10.1038/s41541-022-00526-5, 10.1093/cid/ciac630, 10.1093/ofid/ofac464, 10.1038/s41591-022-01840-0) and this should be mentioned.

Response 3: Unfortunatly we don't have this information and it is difficult to retrospectively find this data.

Comment 4: This study also this needs patient information, consider Table S1 in 10.1016/j.immuni.2021.05.010 as a good way to set this out

Rsponse 4: We added a table with patients information that is table 1.

Comment 5: The fact that the cohort were all hospitalised is a key limitation which should be mentioned in the discussion

Response 5: Done.

comment 6: Line 111 – “Some participants concluded the follow up early due to good general health conditions” – does this mean some reported no symptoms at that time point?

Response 6: No, it does not mean that. it mean that for some patients that for example lived far from the hospital, we didn't program a successive control visit if they don't need a specialistic opinion.

Comment 7: Bias towards men and older population should be mentioned in the discussion

Response 7: Done.

Comment 8: I think it is best to mention that the biomarkers correlate with specific symptoms, not Long COVID in general. In order for the biomarkers to work for Long COVID, a non-symptomatic control is needed in this study.

Response 8: I agree. it has been modified in the discussion.

Comment 9:What this article desperately needs is more connections to the vast Long COVID literature, and discussion of how this works fits into the field as a whole. Many studies have tracked long COVID symptoms over time and the authors should discuss how this work connects to other work in the literature.

Response 9: done.

Reviewer 3 Report

Comments and Suggestions for Authors

Thank you for the opportunity to review the manuscript titled “Severe acute SARS-CoV-2 infection and Long COVID: What Do We Know So Far? New Challenges in Diagnosis and Management.” The authors have conducted an interesting work on a relevant topic that is still being actively studied, namely Long COVID. However, the manuscript presents a series of shortcomings that should be addressed. Below, I list several suggestions to improve the quality of the manuscript.

Introduction

-The introduction is too simple, the first four paragraphs consist of only 2–3 lines each. I suggest expanding the introduction further.

-I recommend better organization of the introduction, as it currently reads like a somewhat disordered list of symptoms without smooth transitions between paragraphs or ideas.

-Lines 36–37: The definition of Long COVID is inadequate. I suggest reviewing the definition provided by the World Health Organization.

-I recommend including a paragraph that discusses the epidemiology of Long COVID, its risk factors, and its global impact.

Methodology

-I suggest organizing the methodology section better by dividing it into subsections for easier reading, including sections such as:

1.Study Design

2.Sample and Setting

3.Inclusion Criteria

4.Exclusion Criteria

5.Statistical Analysis

Results

-Improve the quality of Figures 1, 2, 3, 4, and 6, as they are somewhat difficult to read.

-In Figure 5, the value for the green section is not indicated.

-Line 254: The legend of the figure should say Figure 6 instead of Figure 5.

-For Tables 3 and 4, I suggest providing a more detailed description of the results in the paragraphs citing these tables.

Discussion

-Again, there are too many short paragraphs of 2 to 3 lines. In academic writing, paragraphs should develop complete ideas and be well connected. Excessive fragmentation makes the information read like a list.

-I suggest starting the discussion with an introductory paragraph highlighting the main findings of the study.

-In the discussion, the authors address the topic of Long COVID symptoms; however, they do not present statistical data. For example, the literature reports percentages of patients who have experienced arrhythmias, tinnitus, brain fog, among others. I suggest reviewing the following article: https://www.mdpi.com/1660-4601/19/22/14673

-I recommend a more detailed discussion comparing your findings with other related studies conducted in different countries.

-I suggest including a paragraph that discusses Long COVID in special populations, such as pregnant women. I recommend reviewing the following article: https://pmc.ncbi.nlm.nih.gov/articles/PMC9851727/#sec0010

-The discussion would be enriched by developing a paragraph on the impact of vaccination on Long COVID.

-I suggest developing a paragraph on the public health impact of Long COVID.

Additional comment

 Please include the ethics committee that approved the study, as well as the approval number.

Author Response

Comment 1: The introduction is too simple, the first four paragraphs consist of only 2–3 lines each. I suggest expanding the introduction further.

Response 1: Done.

Comment 2:  recommend better organization of the introduction, as it currently reads like a somewhat disordered list of symptoms without smooth transitions between paragraphs or ideas.

Response 2: done.

Comment 3: -Lines 36–37: The definition of Long COVID is inadequate. I suggest reviewing the definition provided by the World Health Organization.

Response 3: done.

Comment 4: -I recommend including a paragraph that discusses the epidemiology of Long COVID, its risk factors, and its global impact.

 Response 4: done.

Comment 5: I suggest organizing the methodology section better by dividing it into subsections for easier reading, including sections such as:

1.Study Design

2.Sample and Setting

3.Inclusion Criteria

4.Exclusion Criteria

5.Statistical Analysis

Response 5: Done.

Comment 6: Improve the quality of Figures 1, 2, 3, 4, and 6, as they are somewhat difficult to read.

Response 6: Done.

Comment 7: In Figure 5, the value for the green section is not indicated.

Response 7: done.

Comment 8: Line 254: The legend of the figure should say Figure 6 instead of Figure 5.

Response 8: done.

Comment 9: For Tables 3 and 4, I suggest providing a more detailed description of the results in the paragraphs citing these tables.

Response 9: thanks for the suggestion, it would be interesting to expand this description too but I preferred not to do so because many data in this table were not relevant to the discussion. I let the reader see them (if interested) only in the table.

Comment 10: Again, there are too many short paragraphs of 2 to 3 lines. In academic writing, paragraphs should develop complete ideas and be well connected. Excessive fragmentation makes the information read like a list.

response 10: done

Comment 11: I suggest starting the discussion with an introductory paragraph highlighting the main findings of the study.

Response 11: a small introduction is already present, I didn't think it was useful to extend it so as not to create repetitions

Comment 12: In the discussion, the authors address the topic of Long COVID symptoms; however, they do not present statistical data. For example, the literature reports percentages of patients who have experienced arrhythmias, tinnitus, brain fog, among others. I suggest reviewing the following article: https://www.mdpi.com/1660-4601/19/22/14673

Response 12: there are no statistical data on the symptoms because this was not the purpose of the study, but only to do a descriptive analysis. the statistical results are limited to the part on the markers. it would be interesting to do another study focusing on the identification of the risk factors and complete a statistical comparison. thanks for the suggestion.

Comment 13: -I recommend a more detailed discussion comparing your findings with other related studies conducted in different countries.

Response 13: Done.

Comment 14: I suggest including a paragraph that discusses Long COVID in special populations, such as pregnant women. I recommend reviewing the following article: https://pmc.ncbi.nlm.nih.gov/articles/PMC9851727/#sec0010

Response 14: I think it is not relevant as there are no pregnant women in the group

Comment 15: The discussion would be enriched by developing a paragraph on the impact of vaccination on Long COVID.

Response 15: Not having the vaccination data of the patients I have no content to add to the discussion

Comment 16: -I suggest developing a paragraph on the public health impact of Long COVID.

Response 16: It has already been added in the introduction

Comment 17:lease include the ethics committee that approved the study, as well as the approval number.

Response 17: Ethical review and approval were waived for this study due to the local legislation. The work analysis results were obtained exclusively from routine clinical practice in accordance with the clinician’s prescription at the sample time. The clinical management of patients was not affected, and no analysis were performed on the stored samples.

Round 2

Reviewer 1 Report

Comments and Suggestions for Authors

Dear Authors,

thank you for your answers 

good luck!

Author Response

Thank you.

Reviewer 2 Report

Comments and Suggestions for Authors

The symptoms reported in this work (long COVID in hospitalised COVID-19 patients) have been reported countless times since mid 2020. The findings here are consistent with the field but there has been progress since 2020. Particularly, the effects of prior immunity from vaccination, the mechanisms causing different Long COVID symptoms and the trajectories of these symptoms over multiple years. Additionally, the rates of recovery of patients is now a major area of interest. 

Overall, the work is too small in scope for publication in Diseases. 

Author Response

Thank you for advices.

Reviewer 3 Report

Comments and Suggestions for Authors

The authors have properly addressed the indicated comments